# The Predictive Value of Gut Microbiota Composition for Sustained Immunogenicity following Two Doses of CoronaVac

**DOI:** 10.3390/ijms25052583

**Published:** 2024-02-23

**Authors:** Ho-Yu Ng, Yunshi Liao, Ruiqi Zhang, Kwok-Hung Chan, Wai-Pan To, Chun-Him Hui, Wai-Kay Seto, Wai K. Leung, Ivan F. N. Hung, Tommy T. Y. Lam, Ka-Shing Cheung

**Affiliations:** 1School of Clinical Medicine, The University of Hong Kong, Hong Kong; nghoyu@connect.hku.hk; 2Department of Medicine, School of Clinical Medicine, The University of Hong Kong, Queen Mary Hospital, Hong Kong; liaoyunshi@gmail.com (Y.L.); zhangrq@hku.hk (R.Z.); elviswpto@gmail.com (W.-P.T.); hch309@ha.org.hk (C.-H.H.); wkseto@hku.hk (W.-K.S.); waikleung@hku.hk (W.K.L.); ivanhung@hku.hk (I.F.N.H.); 3Department of Microbiology, School of Clinical Medicine, The University of Hong Kong, Queen Mary Hospital, Hong Kong; chankh2@hku.hk; 4State Key Laboratory of Emerging Infectious Diseases, School of Public Health, The University of Hong Kong, Hong Kong; 5Centre for Immunology & Infection Ltd., 17W Hong Kong Science & Technology Parks, Hong Kong; 6Laboratory of Data Discovery for Health Ltd., 19W Hong Kong Science & Technology Parks, Hong Kong; 7School of Public Health, The University of Hong Kong, Hong Kong

**Keywords:** gut microbiota, vaccine, CoronaVac, COVID-19 vaccine, vaccine immunogenicity

## Abstract

CoronaVac immunogenicity decreases with time, and we aimed to investigate whether gut microbiota associate with longer-term immunogenicity of CoronaVac. This was a prospective cohort study recruiting two-dose CoronaVac recipients from three centres in Hong Kong. We collected blood samples at baseline and day 180 after the first dose and used chemiluminescence immunoassay to test for neutralizing antibodies (NAbs) against the receptor-binding domain (RBD) of wild-type SARS-CoV-2 virus. We performed shotgun metagenomic sequencing performed on baseline stool samples. The primary outcome was the NAb seroconversion rate (seropositivity defined as NAb ≥ 15AU/mL) at day 180. Linear discriminant analysis [LDA] effect size analysis was used to identify putative bacterial species and metabolic pathways. A univariate logistic regression model was used to derive the odds ratio (OR) of seropositivity with bacterial species. Of 119 CoronaVac recipients (median age: 53.4 years [IQR: 47.8–61.3]; male: 39 [32.8%]), only 8 (6.7%) remained seropositive at 6 months after vaccination. *Bacteroides uniformis* (log_10_LDA score = 4.39) and *Bacteroides eggerthii* (log_10_LDA score = 3.89) were significantly enriched in seropositive than seronegative participants. Seropositivity was associated with *B. eggerthii* (OR: 5.73; 95% CI: 1.32–29.55; *p* = 0.022) and *B. uniformis* with borderline significance (OR: 3.27; 95% CI: 0.73–14.72; *p* = 0.110). Additionally, *B. uniformis* was positively correlated with most enriched metabolic pathways in seropositive vaccinees, including the superpathway of adenosine nucleotide de novo biosynthesis I (log_10_LDA score = 2.88) and II (log_10_LDA score = 2.91), as well as pathways related to vitamin B biosynthesis, all of which are known to promote immune functions. In conclusion, certain gut bacterial species (*B. eggerthii* and *B. uniformis*) and metabolic pathways were associated with longer-term CoronaVac immunogenicity.

## 1. Introduction

Since early 2020, the COVID-19 pandemic has already affected over 700 million people and caused more than 6 million deaths around the world [1]. Vaccination remains one of the most effective measures in the fight against the pandemic, and over 13 billion doses of COVID-19 vaccines have been administered worldwide [1] to offer protection against SARS-CoV-2 infection, severe symptoms, and death [2]. However, prior studies showed that antibody levels could drastically decline over time [3,4]. Despite this, factors that might influence the durability of vaccine immunogenicity remain uninvestigated.

The gut microbiota was reported to have the ability to modulate immune response towards various vaccines, including oral rotavirus and flu vaccines [5]. This was achieved via various mechanisms, such as the activation of pattern recognition receptors (PRRs) that could regulate antigen-presenting cell (APC) activity [6], as well as the secretion of immunomodulatory metabolites, including flagellin [7], lipopolysaccharides (LPS) [6], peptidoglycans [8], short-chain fatty acids (SCFAs) [9], and secondary bile acids [10]. In a randomized controlled trial, the level of influenza vaccine-induced antibodies was attenuated by antibiotic-induced gut dysbiosis, which also resulted in increased inflammatory signalling and disturbed plasma metabolome [10].

Consistently, evidence of the potential role of the gut microbiota in COVID-19 vaccine immunogenicity is starting to emerge [11]. Pre-vaccination antibiotics have been shown to be associated with lower vaccine immunogenicity as well as higher risk of COVID-19 infection, severe disease outcomes, and hospitalization [12,13]. Two studies which recruited mostly non-immunocompromised subjects have found that several bacterial species were associated with a higher antibody titer at one month after two doses of CoronaVac (inactivated virus vaccine) (e.g., *Bifidobacterium adolescentis*, *Adlercreutzia equolifaciens*, and *Asaccharobacter celatus*) [14] and at 42 days after two doses of BBIBP-CorV (inactivated virus vaccine) (e.g., *Collinsella aerofaciens* and *Fusicatenibacter saccharivorans*) [15]. However, the gut microbiota’s potential role in long-term immunogenicity after COVID-19 vaccination has not been investigated thus far.

Therefore, we conducted this multi-centre, prospective cohort study to investigate the association between the baseline composition of the gut microbiota and the vaccine immunogenicity at 6 months after CoronaVac vaccination.

## 2. Results

### 2.1. Demographics and Baseline Characteristics

A total of 119 eligible adults who received two doses of CoronaVac vaccine were recruited. Thirty-nine (32.8%) of the participants were male, and the median age was 53.4 years (IQR: 47.8–61.3; range 19.7–77.6). The baseline demographics between seropositive and seronegative vaccinees were comparable (all *p* > 0.05; Table 1). At day 180, only eight (6.7%) of the participants remained seropositive (median NAb level: 19.4 AU/mL; IQR: 18.2–22.8), while the rest were seronegative (median NAb level: 5.8 AU/mL; IQR: 4.0–7.6).

Upon univariate logistic regression, there were no clinical factors that were significantly associated with seropositivity at day 180 (all *p* > 0.05; Table 2). Similarly, upon univariate linear regression, there were no clinical factors that were significantly associated with NAb level at day 180 (all *p* > 0.05; Table 2).

### 2.2. Baseline Gut Microbiota Composition Was Associated with Immune Response to CoronaVac at Day 180

We found no significant difference in terms of various alpha diversity measures, including Shannon index, Simpson index, and richness (all *p* > 0.05; Appendix A), as well as beta diversity as indicated by PERMANOVA analysis (*p* = 0.447; Appendix A) between seropositive and seronegative vaccinees. Despite this, LefSe analysis identified 12 bacterial species that were enriched in seropositive vaccinees, as well as one species that was depleted. Among these species, two *Bacteroides* species (*Bacteroides uniformis* and *Bacteroides eggerthii*) were not zero-inflated (i.e., their median relative abundance was not equal to zero). These two species were both associated with seropositivity (*B. uniformis*: log_10_LDA score = 4.39, *p* = 0.025; *B. eggerthii*: log_10_LDA score = 3.89, *p* = 0.023) (Figure 1A) and were consistently found to be significantly more abundant in seropositive vaccinees than in seronegative vaccinees (*B. uniformis*: 9.48% vs. 4.37%, *p* = 0.025; *B. eggerthii*: 0.40% vs. 0%, *p* = 0.023) (Figure 1B). Univariate logistic regression showed that a higher abundance of *B. eggerthii* was significantly associated with seropositivity (OR = 5.73; 95% CI: 1.32–29.55), while *B. uniformis* achieved borderline statistical significance with seropositivity (OR = 3.27; 95% CI: 0.73–14.72; Table 2). During univariate linear regression, *B. eggerthii* was able to remain statistically significant (beta coefficient = 2.52, 95% CI: 0.24–4.79) (Table 2).

### 2.3. Association between Baseline Gut Microbiota and Metabolic Pathways and Vaccine Immunogenicity at Day 180

Additionally, we found that 14 metabolic pathways were enriched in the baseline gut metabolome of seropositive vaccinees, while 6 were depleted (Appendix A). These pathways either belonged to the “Biosynthesis” category, the “Degradation/Utilization/Assimilation” category, or the “Generation of Precursor Metabolites and Energy” category (Appendix A). In seropositive vaccinees, there were a number of enriched pathways related to the de novo synthesis of nucleotides or deoxyribonucleotides, such as the superpathway of adenosine nucleotide de novo biosynthesis I (log_10_LDA score = 2.86; *p* = 0.020) and II (log_10_LDA score = 2.90; *p* = 0.012; Appendix A). There were also pathways related to vitamin B6 synthesis, including the superpathway of pyridoxal 5′-phosphate biosynthesis and salvage (log_10_LDA score = 2.72; *p* = 0.035) and pyridoxal 5-phosphate biosynthesis I (log_10_LDA score = 2.71; *p* = 0.033) (Appendix A). A few enriched pathways in seropositive vaccinees were related to degradation of various substances, such as L-histidine degradation I (log_10_LDA score = 2.84; *p* = 0.002) and III (log_10_LDA score = 2.78; *p* = 0.007) (Appendix A).

### 2.4. Correlation between Gut Microbiota and Metabolic Pathways on Vaccine Immunogenicity

Spearman’s correlation analysis between the baseline gut bacterial species and metabolic pathways found that most of the metabolic pathways enriched in seropositive vaccinees were positively associated with *B. uniformis* (Figure 2). Notable examples included the superpathway of adenosine nucleotide de novo biosynthesis I (Spearman’s r = 0.26; *p* = 0.047) and II (r = 0.29; *p* = 0.018), L-histidine degradation I (r = 0.61; *p* < 0.001) and III (r = 0.65; *p* < 0.001), superpathway of pyridoxal 5′-phosphate biosynthesis and salvage (r = 0.70; *p* < 0.001), and pyridoxal 5-phosphate biosynthesis I (r = 0.68; *p* < 0.001).

### 2.5. Association between Baseline Gut Microbiota Composition and CoronaVac-Related Adverse Events

#### 2.5.1. After 1st Dose Vaccine

Of the 119 participants, 69 (58.0%) subjects reported adverse events after the first dose injection (Appendix A), all of which were mild (grades 1 and 2) and self-limiting. Injection site pain (43.7%) and fatigue (26.9%) were the most common local and systemic adverse events, respectively.

No significant difference in alpha diversity (richness, Shannon, and Simpson index) (all *p* > 0.05; Appendix A) and beta diversity on PERMANOVA analysis (*p* = 0.902; Appendix A) were seen between those with and without adverse reactions. LefSe analysis found that four species were enriched in those who developed adverse effects, while eight species were enriched in those who did not (Appendix A). Non-zero-inflated species included *Bacteroides thetaiotaomicron* (log_10_LDA score = 3.71; 1.02% vs. 0.70%; *p* = 0.044), *Bacteroides cellulosilyticus* (log_10_LDA score = 3.25; 0.13% vs. 0.03; *p* = 0.016), *Clostridium* sp. *CAG 58* (log_10_LDA score = −2.60; 0.006% vs. 0.0%; *p* = 0.047), and *Bacteroides plebeius* (log_10_LDA score = −4.11; 1.52% vs. 0.0%; *p* = 0.046). *B. thetaiotaomicron* and *B. cellulosilyticus* were associated with adverse events after the first dose, while *Clostridium* sp. *CAG 58* and *B. plebeius* were associated with a lower risk of adverse events after the first dose (Appendix A).

#### 2.5.2. After 2nd Dose Vaccine

Data from three participants were missing for the adverse reactions within seven days after the second dose. Of those who responded, 52 (44.8%) subjects reported adverse events (Appendix A). All adverse events were mild (grades 1 and 2) and self-limiting. Same as after the first dose vaccine, injection site pain (32.8%) and fatigue (14.7%) were the most common local and systemic adverse events, respectively.

Again, no significant difference in alpha diversity (richness, Shannon, and Simpson index) (all *p* > 0.05; Appendix A) and beta diversity on PERMANOVA analysis (*p* = 0.187; Appendix A) were seen between those with and without adverse reactions. LefSe analysis found that 15 species were enriched in those who developed adverse effects, while 10 species were enriched in those who did not (Appendix A). A total of 13 species were not zero-inflated, 10 of which were enriched in those who experienced adverse effects and 3 enriched in those who did not. The three species enriched in those without adverse events were *Faecalibacterium prausnitzii* (log_10_LDA score = −4.05; 7.94% vs. 5.89%; *p* = 0.035), *Roseburia intestinalis* (log_10_LDA score = −3.26; 0.24% vs. 0.02%; *p* = 0.014), and *Coprococcus comes* (log_10_LDA score = −2.60; 0.15% vs. 0.04%; *p* = 0.049) (Appendix A). On the other hand, notable species enriched in those with adverse events included *Parabacteroides distasonis* (log_10_LDA score = 3.56; 1.27% vs. 0.81%; *p* = 0.014), *Clostridium bolteae* (log_10_LDA score = 2.79; 0.02% vs. 0.003%; *p* = 0.018), *Clostridium symbiosum* (log_10_LDA score = 2.57; 0.0002% vs. 0.0%; *p* = 0.025), as well as *B. thetaiotaomicron* (which was also enriched in those with adverse effects after first dose) (log_10_LDA score = 3.31; 1.16% vs. 0.57%; *p* < 0.001) (Appendix A).

## 3. Discussion

Our study was the first to demonstrate that the gut microbiota was associated with immunogenicity towards CoronaVac inactivated vaccine at 6 months post-vaccination. We found that *B. uniformis* and *B. eggerthii* served as potential microbial markers that might predict long-term vaccine immunogenicity. Metabolic markers potentially predictive of long-term vaccine immunogenicity were also identified, and we found that most of the metabolic pathways enriched in seropositive vaccinees were positively correlated with *B. uniformis*.

The gut microbiota, via its ability to produce immunomodulatory metabolites, is involved in modulating the body’s immune response towards vaccinations. Examples included flagellin, lipopolysaccharide (LPS) and peptidoglycan, which influence the immune system via activating pattern recognition receptors (PRRs), including Toll-like receptors (TLRs) or NOD-like receptors. Flagellin could improve influenza vaccine immunogenicity via TLR5-mediated enhancement of short-lived plasma cell presence [7]. Lipopolysaccharide and peptidoglycan can be sensed by TLR-4 and NOD2 receptor, respectively, with the former boosting antibody production and type 1 T helper cells (Th1) and the latter involved in the mucosal adjuvant activity of cholera toxin [6,8]. The gut microbiota could also secrete other metabolites, including general nutrients such as amino acids and vitamins, as well as microbiota-specific substances such as SCFAs and secondary bile acids. Previous studies have shown that SCFAs can increase energy production in B cells via different means, thereby promoting antibody production [9], whereas secondary bile acids were negatively correlated with inflammatory markers after influenza vaccination [10].

Studies conducted on non-immunocompromised adults have shown that the gut microbiota was associated with vaccine immunogenicity after COVID-19 vaccination [14,15]. One study also conducted in Hong Kong measured CoronaVac immunogenicity using the SARS-CoV-2 surrogate virus neutralisation test (sVNT). In this study, the median age of the CoronaVac vaccinees was 55 years old, which was similar to that in our study (53.4 years old). This study found that while CoronaVac only achieved sVNT of 57.6% at one month after the second dose of vaccination, *B. adolescentis* was enriched in the baseline gut microbiome of CoronaVac high responders (defined as sVNT inhibition > 60%), and it remained persistently high at one month after the second dose of vaccination. *B. adolescentis* was positively correlated with sVNT% and pathways related to carbohydrate metabolism, which were also enriched in high responders [14]. On the other hand, low responders had a lower baseline abundance of *Bacteroides vulgatus*, *B. thetaiotaomicron,* and *Ruminococcus gnavus*, and *B. vulgatus* remained persistently low at one month after the second dose of vaccination [14]. Another study conducted in China also found that there was enrichment of *C. aerofaciens, F. saccharivorans, Eubacterium ramulus,* and *Veillonella dispar* in the baseline microbiome in high responders (defined as top 25% of angiotensin-converting enzyme 2-receptor-binding domain (ACE2-RBD) inhibiting antibody concentrations) at day 14 after second dose of BBIBP-CorV inactivated SARS-CoV-2 vaccine, while *Lawsonibacter asaccharolyticus* was enriched in low responders at baseline [15]. Of note, these studies only studied the association between gut microbiota and vaccine immunogenicity up to one month after the second dose. Our present study investigated vaccine immunogenicity at six months after the second dose and found that *B. uniformis* and *B. eggerthii* were enriched in the baseline microbiome in high responders.

Previous literature had shown that *B. uniformis* and *B. eggerthii,* which we identified in the current study, were able to boost the immune system (Figure 3). *B. uniformis* can restore immune dysfunction in obese mice by enhancing the ability of macrophages and dendritic cells (DCs) to produce cytokines, as well as restoring the capacity of DCs in antigen presentation and stimulation of T cell proliferation [16]. When combined with wheat bran extract, its preferred carbon source, *B. uniformis* can restore the proportion of induced intraepithelial lymphocytes and type-3 innate lymphoid cells in the intestinal epithelium, as well as produce SCFAs [17]. *B. eggerthii* can produce propionate, another SCFA, in a vitamin B12-dependent manner [18]. In mouse models, *B. eggerthii* was shown to enhance colitis, suggesting pro-inflammatory abilities [19].

In addition, *B. uniformis* was positively correlated with pathways involved in the de novo synthesis of various purines, nucleotides and deoxynucleotides. Purinergic signalling serves as one of the linkages between gut microbiota and immune function [20]. Extracellular adenosine triphosphate (ATP) produced from these pathways can promote the maturation of DCs and stimulation of B and T cells by activating the P2X receptor [20,21,22]. *B. uniformis* was also significantly correlated with pathways that produced beneficial products. These include L-histidine degradation I and III pathways, which can produce L-glutamate as the product [23]. Glutamate plays an important role in protecting T cells from antigen-induced apoptotic cell death, as well as in key cell functions, including adhesion, migration and proliferation [24]. Also included were two pathways related to pyridoxal 5-phosphate (vitamin B6) synthesis, which were the superpathway of pyridoxal 5′-phosphate biosynthesis and salvage as well as pyridoxal 5-phosphate biosynthesis I pathway. Vitamin B6 can improve immune functions by enhancing antibody production, interactions between cytokines and chemokines as well as lymphocyte migration from lymphoid organs to peripheral sites [25,26]. In gastric cancer patients, supplementation with a vitamin B mixture containing vitamin B6 was able to bring profound improvement in lymphocyte counts and functioning [27], further supporting their beneficial roles in boosting immune functions.

Moreover, we observed that different bacterial species were associated with the occurrence of adverse effects after the first or second dose of vaccination. Notably, *B. thetaiotaomicron* was the only species that was enriched in participants who developed adverse effects both after the first dose and second dose. *B. thetaiotaomicron* can exert pro-inflammatory effects by producing LPS, which in turn increases the numbers of transcripts of the pro-inflammatory cytokine IL-6 as well as type 1 IFN-dependent marker IFIT1 via TLR-4 mediated-signalling [28]. Studies have also shown that *B. thetaiotaomicron* can secrete outer membrane vesicles (OMVs) with bacterial sulfatase activity, thereby enabling antigens to access host immune cells and inducing colitis in mouse models [29]. In contrast, species enriched in those without adverse effects, notably *F. prausnitizii*, *R. intestinalis* and *C. comes* had anti-inflammatory properties. These three bacterial genera were known SCFA producers, and SCFA can exert anti-inflammatory effects by inhibiting the action of histone deacetylase as well as the nuclear factor-κB (NF-κB) pathway, which produces pro-inflammatory cytokines in immune cells [30]. Indeed, these three genera had been reported to be depleted in COVID-19 patients [11] as well as in inflammatory bowel disease (IBD) patients [31,32,33], hence further supporting their role in alleviating inflammation. Additionally, SCFAs are involved in regulating energy, fat homeostasis, and the blood–brain barrier, therefore having a role to play against fatigue [34], which was the most common post-vaccination systemic side effect in our study.

Our study has several limitations. First, the sample size was relatively small, with a limited number of patients remaining seropositive at 6 months post-vaccination or developing adverse effects after each dose of vaccination. This may cause this study to be underpowered to detect other gut microbial species that are potentially associated with vaccine immunogenicity and adverse effects development as well. Second, our study findings were not validated on a validation cohort. Third, a lack of animal model study means that the causal relationship between the bacterial species, metabolic pathways and vaccine immunogenicity is not definitively proven. In addition, there was a lack of metabolomics data to support the association between the enriched metabolic pathways with higher vaccine immunogenicity. There was also a lack of gut microbiota data at other time points to investigate if changes in the gut microbiota might have influenced seroconversion at day 180. Dietary data were also unavailable which might also have influenced gut microbiota composition over time. Moreover, our findings may not be generalizable to other populations, as gut microbiota composition varies greatly across different populations and geographical regions due to factors including diet, lifestyle and socioeconomic status [5]. Last, as most of the participants received a third booster dose after six months, it was not possible to conduct a longer follow-up study on the vaccine response durability after two doses of vaccine. Nonetheless, we are planning to further investigate whether vaccine immunogenicity and durability can be predicted by the gut microbiota after receiving booster doses.

## 4. Methods

### 4.1. Study Design and Participants

We conducted a multi-centre, prospective cohort study. Adult subjects who received two doses of CoronaVac vaccines were recruited from three vaccination centres in Hong Kong (Sun Yat Sen Memorial Park Sports Centre, Ap Lei Chau HKU Vaccination Centre, and Queen Mary Hospital). Exclusion criteria included age <18 years, inflammatory bowel disease, immunocompromised status including the use of immunosuppressive agents/chemotherapy and post-transplantation, other medical conditions including rheumatological and autoimmune diseases, cancer, and hematological diseases, and prior COVID-19 infection. Prior COVID-19 infection was identified from history taking or by the presence of SARS-CoV-2 nucleocapsid (N) protein-specific antibodies measured by using a chemiluminescence immunoassay kit (Shenzhen YHLO Biotech), which was shown to have a sensitivity of 77% and specificity of 100% [35]. CoronaVac vaccines were poor inducers of SARS-CoV-2 nucleocapsid (N) protein-specific antibodies; hence, these antibodies can be used to indicate past SARS-CoV-2 infection [36]. This study was approved by the Institutional Review Board (IRB) of the University of Hong Kong and Hospital Authority Hong Kong West Cluster (UW 21-216). All participants provided written informed consent for participation in this study.

### 4.2. Collection of Demographics, Blood and Stool Samples

Basic demographics, including sex, age, antibiotics [12], the use of proton pump inhibitors (PPIs), and prebiotics and/or probiotics within six months before vaccination were collected. Subjects received two doses of intramuscular CoronaVac (0.5 mL) 4 weeks apart. We collected blood samples (i) before vaccination (baseline) and (ii) at 180 days after the first dose, as a significant decline in antibody levels at 6 months after vaccination with CoronaVac has been observed [3,4]. We also collected baseline stool samples using the OMNIgene tube before vaccine administration and stored them at −80 °C until DNA extraction. Total genomic DNA from baseline stool samples was extracted, followed by shotgun metagenomic sequencing using the Illumina NovaSeq 6000 platform (Illumina, San Diego, CA, USA). Detailed steps can be found in Appendix A.

Vaccine immunogenicity was measured as the level of neutralizing antibodies (NAbs) against the SARS-CoV-2 receptor-binding domain (RBD), as it was shown to be a suitable surrogate marker of vaccine effectiveness [37] that could predict protection from symptomatic COVID-19 infection [38,39]. The new version of the iFlash-2019-nCoV NAb kit (chemiluminescent microparticle immunoassay; Shenzhen YHLO Biotech Co, Ltd., Shenzhen, China) was used to test for this NAb [40]. Detailed steps can be found in the Appendix A.

### 4.3. Primary and Secondary Outcomes of Interest

The primary outcome of interest was the NAb seroconversion rate at day 180, with NAb ≥ 15 AU/mL defined as being seropositive.

The secondary outcome was any adverse reactions recorded by the participants daily for 7 days after each dose of vaccination. These include local reactions (pain, erythema, swelling, ecchymosis, and itchiness) as well as systemic reactions (fever, chills, headache, tiredness, nausea, vomiting, diarrhea, myalgia, arthralgia, and skin rash). The severity of adverse reactions was graded as 1 to 4 according to the toxicity grading scale by the U.S. Department of Health and Human Services [41].

### 4.4. Statistical Analysis

We conducted all statistical analyses using R version 4.2.2 (R Foundation for Statistical Computing, Vienna, Austria) statistical software. We used the Mann–Whitney U test for two continuous variables and the Chi-square test or Fisher exact test for categorical variables to assess the statistical significance between groups, respectively. A univariate logistic and linear regression model was used to estimate the odds ratio (OR) and beta coefficients of seropositivity and NAb level with different variables, respectively. False discovery rate (FDR) was used to correct for multiple comparisons in multiple hypothesis testing.

Raw NGS reads were processed by fastp (v0.20.1) [42] to quality and adapter trimming to remove sequencing adapters and bases with poor quality. Trimmed reads were subjected to host sequence removal by using Bowtie2 (v2.4.5) [43] to map reads against human reference genome GRCh38.p13. The composition of microbial communities at the species level and the functional profile in each sample were inferred using MetaPhlAn (v3.0) [44] and HUMAnN (v3.0) [45], respectively. Estimation of species coverage and relative abundance was then determined. Alpha diversity in terms of observed species richness, Shannon and Simpson index was calculated using *vegan* package (v2.6.4) in RStudio and compared between groups using Wilcoxon signed-rank test. Beta diversity in terms of Bray–Curtis compositional dissimilarity was compared using non-metric multidimensional scaling (NMDS). Permutational multivariate analysis of variance (PERMANOVA) was then used to compare microbial communities of different samples. LefSe (linear discriminant analysis effect size) was used to identify the putative gut bacterial species and metabolic pathways with an absolute value of linear discriminant analysis (LDA) score ≥2. The top 75% was used to define a high relative abundance of a particular bacterial species. The ORs and the beta coefficient of seropositivity with a high relative abundance of the putative gut bacterial species were determined using a univariate logistic and linear regression model, respectively.

A two-sided *p*-value ≤ 0.05 was considered statistically significant, while a *p*-value between 0.05 and 0.15 was considered borderline significant.

## 5. Conclusions

Certain bacterial species (*B. eggerthii* and *B. uniformis*) could predict CoronaVac immunogenicity at six months. These results may contribute to future research on using gut microbiota as an intervention to enhance the longer-term durability of immune responses towards vaccination.

## Figures and Tables

**Figure 1 ijms-25-02583-f001:**
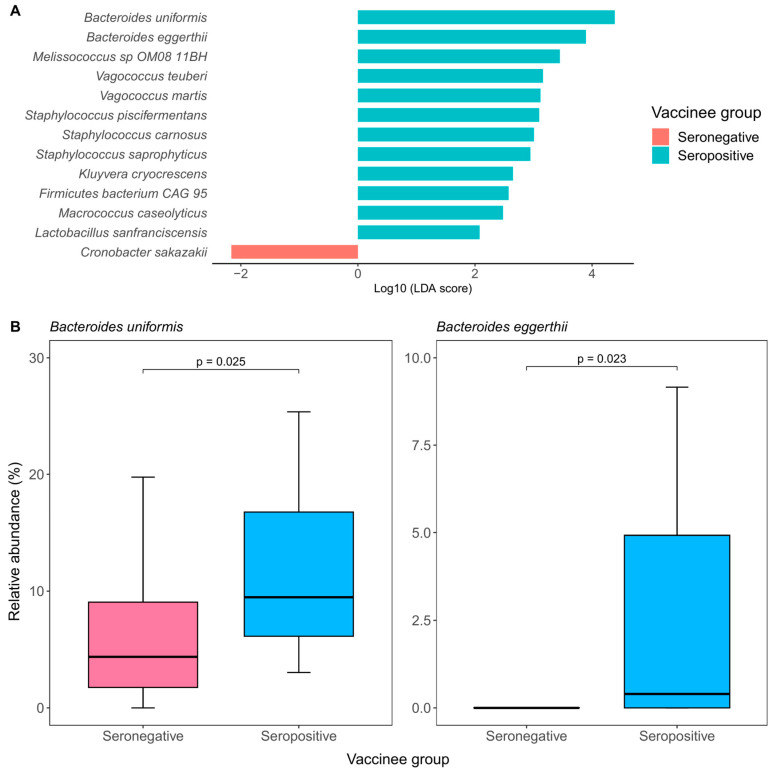
Baseline gut bacterial species enriched in seropositive vs. seronegative vaccinees of CoronaVac at day 180 and their relative abundances. (**A**): Differential gut bacterial species were detected by linear discriminant analysis effect size (LefSe). (**B**): Relative abundances of baseline gut bacterial species among seropositive and seronegative vaccinees of CoronaVac at day 180.

**Figure 2 ijms-25-02583-f002:**
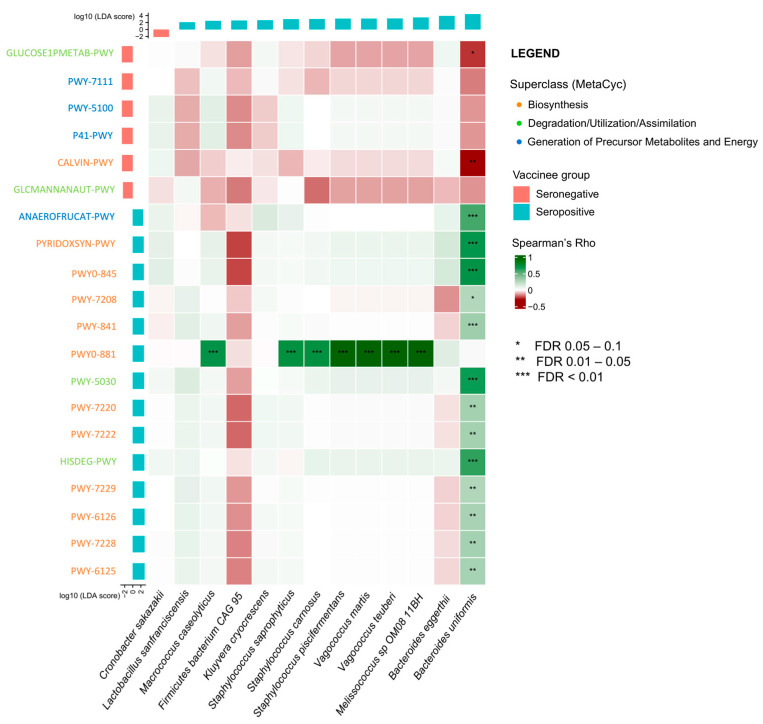
Correlation heatmap showing spearman correlation of baseline metabolic pathways with gut bacterial species between seropositive and seronegative vaccinees of CoronaVac. Putative baseline gut bacterial species and metabolic pathways were detected by LefSe. MetaCyc superclass of each metabolic pathway is denoted by the font colour. Spearman correlations between gut bacterial species and metabolic pathways identified by LefSe were shown, with the FDR value range denoted by a number of asterisks. Abbreviations: FDR, false discovery rate.

**Figure 3 ijms-25-02583-f003:**
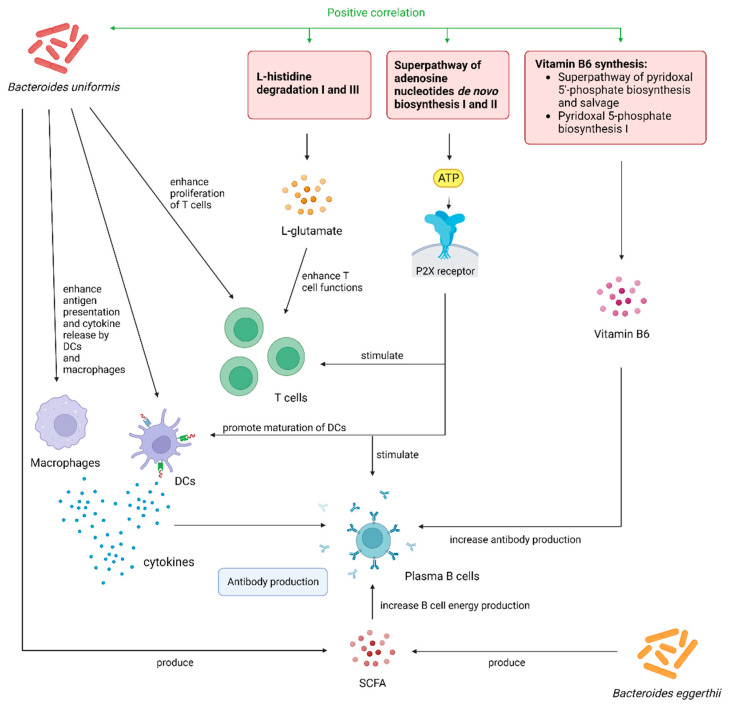
Graphical illustration of mechanistic links between gut microbiota species and metabolic pathways with CoronaVac immunogenicity based on our current findings and prior literature. Bacteroides uniformis can enhance antigen presentation and cytokine release by macrophages and dendritic cells (DCs), as well as the proliferation of T cells. It can also produce short-chain fatty acids (SCFAs), which increase energy production in plasma B cells, hence enhancing antibody production. Additionally, it is positively correlated with L-histidine degradation I and III pathways, which produce L-glutamate, superpathway of adenosine nucleotides de novo biosynthesis I and II pathways, which can produce ATP, as well as pathways related to vitamin B6 biosynthesis. ATP, via binding to P2X receptors and activating purinergic signalling pathways, can stimulate T cells. L-glutamate can enhance T cell functions. Vitamin B6 can increase antibody production and boost lymphocyte functioning. Bacteroides eggerthii, similar to B. uniformis, can also produce SCFAs for enhancing antibody production. Abbreviations: DCs, dendritic cells; ATP, adenosine triphosphate; SCFAs, short-chain fatty acids.

**Table 1 ijms-25-02583-t001:** Baseline characteristics between CoronaVac vaccine recipients with seropositivity and seronegativity at 6 months after vaccination.

Characteristic	Overall, *n* = 119 ^1^	Seronegative, *n* = 111 ^1^	Seropositive, *n* = 8 ^1^	*p*-Value ^2^
Age (years) ≥ 55 (*n*, %)	51 (42.9%)	46 (41.4%)	5 (62.5%)	0.286
Male sex (*n*, %)	39 (32.8%)	39 (35.1%)	0 (0.0%)	0.052
Antibiotics * (*n*, %)	5 (4.2%)	4 (3.6%)	1 (12.5%)	0.298
Proton pump inhibitors * (*n*, %)	14 (11.8%)	13 (11.7%)	1 (12.5%)	>0.999
Prebiotics/Probiotics (*n*, %)	3 (2.5%)	3 (2.7%)	0 (0.0%)	>0.999

^1^ Median (IQR); n (%) ^2^ Wilcoxon rank sum test; Fisher’s exact test * Any use within six months before vaccination.

**Table 2 ijms-25-02583-t002:** Clinical factor and high relative abundance of certain baseline gut microbiota species associated with seropositivity (univariate logistic and linear regression).

	Univariate Logistic Regression			Univariate Linear Regression		
**Characteristic**	**OR ^1^**	**95% CI ^1^**	***p*-Value**	**Beta**	**95% CI ^1^**	***p*-Value**
*Clinical factors*						
Age ≥ 55 years old	2.36	0.55, 11.94	0.257	1.68	−0.33, 3.70	0.101
Sex	NA *	NA *	0.992	−2.03	−4.14, 0.09	0.060
Antibiotics	3.82	0.18, 30.70	0.258	1.21	−3.81, 6.23	0.635
Proton pump inhibitors	1.08	0.06, 6.78	0.947	−0.83	−3.96, 2.29	0.598
*High relative abundance of baseline gut microbiota species*						
*Bacteroides uniformis*	3.27	0.73, 14.72	0.110	0.69	−1.63, 3.01	0.556
*Bacteroides eggerthii*	5.73	1.32, 29.55	0.022	2.52	0.24, 4.79	0.031

^1^ OR = odds ratio, CI = confidence interval, NA *: data were not computable.

## Data Availability

All sequencing data generated in this study have been deposited in the Sequence Read Archive under BioProject PRJNA983760.

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
