# Peer review of "The Predictive Value of Gut Microbiota Composition for Sustained Immunogenicity following Two Doses of CoronaVac"

_ijms, 2024, doi:10.3390/ijms25052583_

Round 1

Reviewer 1 Report

Comments and Suggestions for Authors

The researchers delved into exploring the connection between the initial makeup of the gut microbiota and the immune response generated by the CoronaVac vaccine six months post-vaccination. On the whole, the study is clearly laid out and structured, showcasing a well-thought-out design. Nevertheless, there are areas that may benefit from some refinement and enhancement. I have provided some concerns below:

Lines 88 to 90: Should rewrite again in order to avoid confusion

Line 108 to 100: Based on what statistical metric abundance of B. uniformis reached borderline with seropositivity? There is no definition regarding this issue.

The relevant data for the discussed point is not found in Table 2. Please refer to the correct table or section.

Line 109 to 110: The relevant data for the discussed point is not found in Table 2. Please refer to the correct table or section.

In all of the manuscript: Make slight punctuation and grammatical adjustments for clarity and consistency. For example, in line 331, after “could” it should be a verb. In line 349, the adverb should come at the end of the sentence with a comma before it.

In all of the manuscript: Please ensure conformity to the formal conventions of academic writing. For instance, in lines 163 and 166, using the word “again” is not academic.

Line 145: Wrong percentage, 69 out of 119 is 57.9% or approximately 58%.

Line 153 to 158: Please revise this section to improve the clarity of your statement and consider breaking it down into smaller sentences.

Line 179: Please mention the full genus name “Bacteroides” in B. thetaiotaomicron, as it is mentioned the first time in your manuscript.

Line 307 to 310: Maintain consistency in the format. For example, you could start each criterion with a verb or use a consistent structure throughout the list.

Why were blood samples were collected after 180 days of the first dose of vaccination?

Please provide the reason for the selected number of days.

Comments on the Quality of English Language

Moderate editing of the English language is required.

Reviewer 2 Report

Comments and Suggestions for Authors

In this paper, the Authors use a rigorous statistic approach to determine the association of gut bacterial species and metabolic pathways with long-term CoronaVac immunogenicity in a multi-centre study.

The paper is well written and focused on an important topic. However, this reviewer recommends some moderate adjustments to improve the manuscript:

Major issues

The Authors attempt to demonstrate that gut bacterial species at baseline may have determined/influenced long-term CoronaVac immunogenicity, making gut microbiota appear as a static frame. Instead, an interplay between vaccination and gut microbiota has largely been demonstrated, with vaccination altering gut microbiota composition (see for example 10.3748/wjg.v28.i40.5801, 10.1038/s41392-023-01629-8, and 10.1038/s41541-023-00627-9). In order to confirm that baseline gut bacterial species are true predictors of durable (6-month) CoronaVac immunogenicity, and exclude that changes in gut microbiota and/or microbiome may account for observed persistence of seropositivity in vaccine recipients, statistical analysis should be extended including changes in gut microbiota and microbiome in responders and non-responders patients recorded during the 180 days between the completion of vaccination and seropositivity assessment.  

The pieces of information included in Table 1 and Section 4.1 should be integrated including details about assumption of prebiotics and probiotics that may alter gut microbiota before vaccination, during the period between the administration of the first and second vaccine dose, and after the second dose up to the assessment of CoronaVac long-term immunogenicity (6-month period). Also, patients’ adherence to particular dietary patterns (like vegetarianism) should also be reported. Pre/probiotics and dietary patterns should be included in the regression analysis (for the rationale behind this comment, see for example 10.3390/vaccines11101609). If assumption of prebiotics and/or probiotics occurred, the Authors should explain how they rule out the chance that prebiotics/probiotics altered the obtained results in the Discussion (for example, changing gut microbiota composition, in turn altering immune responses towards vaccination). A similar dissertation should be done in the case of dietary pattern influencing gut microbiota composition.

Minor issues

Page 2 lines 71-72 please remove the comma between subject and verb “However, the gut microbiota’s potential role in long-term immunogenicity after COVID-19 vaccination, has not been investigated thus far.”

Reviewer 3 Report

Comments and Suggestions for Authors

The study is conducted based on the serological status after Covid vaccination. However, the data is not provided.

The sample size is very small, and this raises concerns about the statistical power of the analyses.

Providing metabolic pathway information is not sufficient to prove that the microbiota is modulating the immune response. It is important to verify if metabolomics data from the samples correlate with pathways.

The figures are not given properly. For example, there is no proper legend or description on Fig 2. It is hard to understand what the data is saying.

For reviewing this paper, I read some recent research related to this area. The study “gutSMASH predicts specialized primary metabolic pathways from the human gut microbiota” is an excellent model for figures in this area of research.

Association of microbiota with severity of adverse events post-vaccination is described in the paper. Injection site pain and fatigue are the most common local and systemic adverse events described respectively. How does microbiota relate to these body responses?

Comments on the Quality of English Language

Careful editing of English language is required.

Round 2

Reviewer 2 Report

Comments and Suggestions for Authors

This reviewer would like to thank the Authors for addressing the highlighted issues.

Reviewer 3 Report

Comments and Suggestions for Authors

There is a remarkable improvement after first round of revision. 

Thank you!

Comments on the Quality of English Language

Please check the grammar and spelling thoroughly.

In Figure S4: Abbreviations,

correct to : NAb, neutralising antibody